# OpenReview forum: "Steering Without Side Effects: Improving Post-Deployment Control of Language Models"
_NeurIPS.cc/2024/Workshop/SafeGenAi — SafeGenAi Poster_

### Official Review · Reviewer_15jG · 2024-10-09
**Good tradeoff between model performance and safety**

**Rating:** 7
**Confidence:** 4

**Review:**

This paper proposes a KL-then-steer method, which could retain model performance when increasing model safety. This method provides a better tradeoff than previous approaches. It is better to extend the approach to more model capability/safety measurement.

---

### Official Review · Reviewer_5pvS · 2024-10-09
**Practical relevance of AI safety, particularly in mitigating harmful behaviors in language models post-deployment is a significant step forward in developing safer and more reliable AI systems.**

**Rating:** 7
**Confidence:** 4

**Review:**

**Evaluation Criteria**

**Quality** - The proposed KTS technique of language model control post-deployment is explained in a well structured manner supported by adequate experimental framework focusing on quantifiable evidence.

**Clarity** - The verbiage used in the paper is clear and backed with appropriate visual aids where applicable. Sections on KTS etc is dense especially for readers who are not very well versed with the subject matter area.

**Originality** - The post deployment intervention management using steering vectors combined with KL-divergence minimization appears to be novel especially the specific application around preserving model's capability seems innovative.

**Significance** - With the increasing integration of these models into the real-world application, the proposed idea of controlling and mitigating the unintended behaviors of language models post-deployment seems very crucial.

**Pros:**

1. KL-then-steer (KTS) technique of handling unintended behavior after post deployment is aiming for a balance between performance and safety.
2. The paper outlines a practical method that can be easily adapted to real-world use cases without significant overheads.
3.Dynamic AI systems can make use of the prompt classification integration to help guide when to apply steering vectors.

**Cons:**

1. Mathematical implications of  KL-then-steer technique can be difficult for readers to grasp without subject matter expertise.
2. Prompt classification can add computational complexity.